# Association between midday napping and long-term trajectories of cognitive function among middle-aged and older Chinese adults

Jinghong Huang[1], Dongrui Peng[1], Yutong Zhang[2], Yanan Zhang[1], Xiaohui Wang[1]*

**1** School of Public Health, Lanzhou University, Lanzhou, China, **2** The Second Clinical Medical School, Lanzhou University, Lanzhou, China

\* wangxiaohui@lzu.edu.cn

## Abstract

### Background

The prevalence of dementia has become an increasingly important public health priority. This study investigated the association between midday napping and long-term trajectories of cognitive function in middle-aged and older Chinese adults.

### Methods

Among 4648 participants aged 45+ years extracted from the China Health and Retirement Longitudinal Study (CHARLS). The components of the Telephone Interview of Cognitive Status battery (TICS-10) was used to assess cognitive function. Group-based trajectory modelling (GBTM) was used to identify long-term trajectories of cognitive function. Multinomial logistic regression model was used to estimate risk ratios (RRs) and 95% confidence intervals (CIs).

### Results

Three distinct long-term trajectories of cognitive function reflected patterns of rapid decline, slow decline, and stable. The RR (95% CI) for rapid decline was 1.45 (1.05–2.01) for 0 minutes, 1.49 (1.05–2.12) for 31–90 minutes, and 2.19 (1.41–3.42) for >90 minutes compared with midday napping 1–30 minutes. The RR (95% CI) for slow decline was 1.22 (1.02–1.47) for 0 minutes, 1.27 (1.04–1.55) for 31–90 minutes, and 1.80 (1.38–2.35) for >90 minutes compared with midday napping 1–30 minutes. In addition, the increased risk of cognitive decline that transferred from >90 to 31–90 minutes, switched from 31–90 to >90 minutes, and persisted in >90 minutes compared with midday napping 1–30 minutes, especially rapid decline.

### Conclusions

There was a longitudinal association between no and long (>30 minutes) midday napping and long-term trajectories of cognitive decline, especially rapid decline. The

**Data availability statement:** Data are from the China Health and Retirement Longitudinal Study(CHARLS) and are publicly available from the CHARLS website: https://charls.charlsdata.com/pages/data/111/zh-cn.html. Dataset titles used in this study were 2015 CHARLS Wave 3, 2013 CHARLS Wave2, and 2011 CHARLS Wave1). The present study's authors had no special privileges in accessing these datasets which other interested researchers would not have.

**Funding:** This study was supported by the National Natural Science Foundation of China (Grant No. 72274088). The funders had no role in study design, data collection and analysis, decision to publish, or preparation of the manuscript.

**Competing interests:** The authors have declared that no competing interests exist.

study is a 4-year observational in nature and provides limited evidence for establishing causal relationships.

## Introduction

Dementia as a brain disease is classified as a neurocognitive disorder with multiple forms or subcategories [1,2]. Currently, more than 55 million people worldwide are affected by dementia and this number will reach 131.5 million by 2050 [3]. Approximately 10% of the population will develop dementia during their lifetime [4]. In China, about 3–5% of adults suffer from dementia [5], and the prevalence rate is about 6.0% (adjusted for age and sex) among people aged 60+ years, which is about 15.07 million [6]. The global economic burden of dementia is estimated to be $1.3 trillion in 2019 and will reach $1.7 trillion by 2050 [3]. Subjective cognitive decline is often considered a precursor to dementia and is an important juncture in preventing dementia [1,7]. The working population has a higher number of disability-adjusted life years due to dementia, with a greater physical, psychological, and economic impact on families and society [8,9]. Older adults often experience cognitive decline associated with changes in living environment and lifestyle after retirement [10]. Studies have found that subjective cognitive decline independently predicts dementia and that early intervention can attenuate progression to dementia [11,12].

Midday napping is considered a healthy lifestyle with intervening properties. The prevalence of habitual midday napping ranges from 22% to 69% among older adults worldwide [13–16]. Meanwhile, midday napping is also prevalent in China, as Chinese culture considers it an important part of healthy living [15]. Subjective cognition and dementia represent an urgent issue that has become a public health priority that needs to be addressed, especially among middle-aged and older adults [17,18]. However, despite the importance of this issue, dementia and mild cognitive impairment rarely receive the attention they deserve in China [6]. A nationwide cross-sectional study of 46,011 adults aged 60+ found that 71.4% of dementia patients had never been to a doctor for dementia, and 75.31% of caregivers of dementia patients did not know what dementia was or how to manage it [6]. Therefore, identifying potentially intervenable factors of cognitive decline to develop effective preventive and therapeutic measures is extremely important in the prevention and treatment of dementia in middle-aged and older adults.

Previous reviews and meta-analyses have shown significant associations between midday napping and cognitive decline and dementia [18–20]. However, the evidence for the association between midday napping and cognitive function is inconsistent, with some studies reporting that no and long (e.g., >60 and >90 minutes) midday napping is detrimental to cognitive function [20–22], and others reporting that short midday napping is beneficial [18,19]. Importantly, while previous CHARLS-based studies have explored the relationship between napping and cognitive function both cross-sectional [15,23–25] and longitudinal [21], these studies exhibit some methodological limitations. The cross-sectional evidence is

limited by the design, and long-term effects are needed to be identified. Limited longitudinal research predominantly relied on the linear mixed-effect model. Although the model accounts for variability at single time point cognitive function and time trends across individuals, they may not capture the full range of cognitive development trajectories within different subgroups. In contrast, the group-based trajectory model provides a method to estimate probabilities for multiple developmental trajectories simultaneously, rather than merely modeling the average effect for the study population [26]. This approach is particularly useful for identifying distinct long-term patterns of cognitive function, which is crucial given that cognitive trajectories in middle-aged and older adults can vary significantly over time. For example, in a study of body mass index and trajectories of cognitive function in 5693 Chinese middle-aged and older adults (aged 45 years+), it was found that individuals tended to show the following three trajectories of cognitive function, including rapid decline, slow decline, and stable [27]. Moreover, a few studies have reported that midday napping tends to affect circadian rhythms [28–30]. The time of midday napping affects nighttime sleep, so that midday napping affects the speed of falling asleep and nighttime sleep, and further disturbs daily activities the next day [31]. Hence, it is crucial to investigate the relationship between midday napping and long-term trajectories of cognitive function.

This current study hypothesized that midday napping and its changes in midday napping are strongly associated with the long-term trajectory of cognitive function in middle-aged and older Chinese adults. To test these hypotheses, a longitudinal study was conducted using self-reported information on midday napping and cognitive function from the China Health and Retirement Longitudinal Study (CHARLS). Given that midday napping and cognitive function are dynamic and have large individual heterogeneity [32,33], to explore the association between midday napping and its changes and long-term trajectories of cognitive function is worthwhile.

## Methods

### Study design and participants

This study was a secondary analysis based on CHARLS, a national population-based observational survey conducted across 150 countries of 28 provinces in China [34]. CHARLS includes participants aged 45+ years, with follow-ups every 2–3 years (2013, 2015, 2018, and 2020) from baseline in 2011. However, the orientation and attention questions of cognitive function were changed in 2018. Therefore, our study included only three waves (2011, 2013, 2015) to ensure the reliability and accuracy of repeated measures. The baseline survey was conducted in 2011/06, and the end of the recruitment was 2015/09. Several exclusion criteria were formulated: (1) age less than 45 years (n=368), (2) no complete information on cognitive function (n=7763) (3) cognitive score less than or equal to the mean (12.92568) at baseline (n=3973) to reduce concern about reverse causality, i.e., individuals with cognitive impairment may also exhibit sleep disorders [35], (4) no information on midday napping (n=6), and (5) fewer than 1 repeated assessment of cognitive function from wave 2 to 3 (n=947). Consequently, 4648 participants remained for analysis (Fig 1).

The CHARLS study was approved by the Biomedical Ethics Review Committee of Peking University (IRB00001052–11015). All participants provided written informed consent to participate in the study. The research methods employed in this study adhered to the guidelines outlined in the Declaration of Helsinki, ensuring that ethical considerations were followed.

### Assessment of midday napping

The independent variable in this study was baseline (2011) midday napping, which was self-reported. **Midday napping** was assessed by the following question: "During the past month, how long did you take a nap after launch?" and further classified as: 0, 1–30, 31–90, and >90 minutes. Based on prior epidemiological studies, 30 minutes is a common time for a short midday napping [15,36].

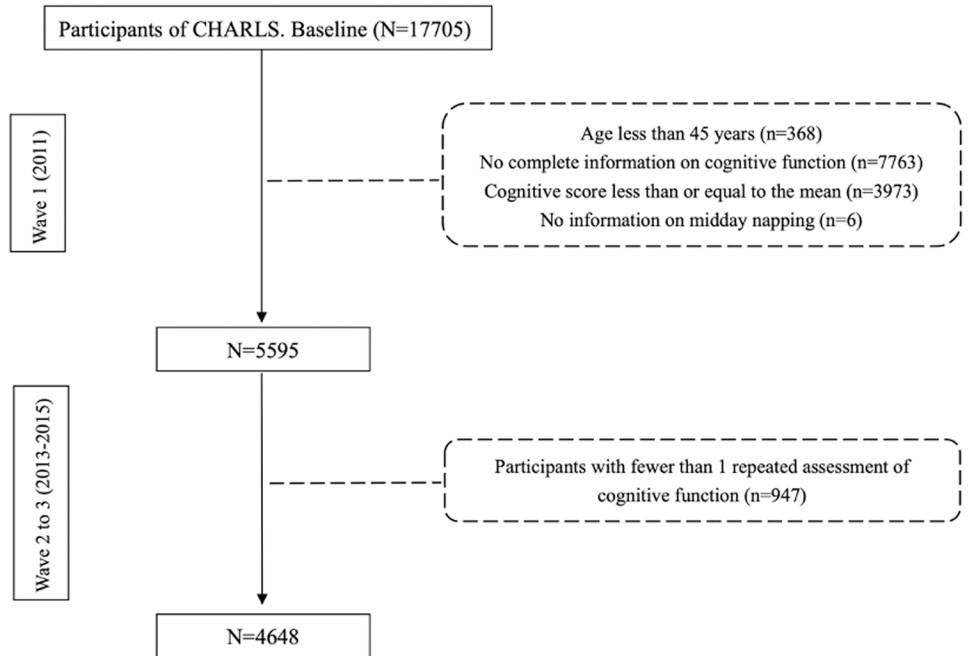

**Fig 1. Flowchart of the study population.**

## Assessment of cognitive function

The dependent variable in this study was long-term trajectories of cognitive function. Cognitive function represented the three ability assessments: orientation, attention, episodic memory, and visual-spatial, measured at baseline (2011) and at 2 subsequent follow-up visits (2013 and 2015). The cognitive function measures in CHARLS were derived from the components of the **Telephone Interview for Cognitive Status battery (TICS-10)**, which specifically assessed orientation and attention [37]. Participants were asked to make five calculations (subtract 7 from 100 consecutively), answer the date (year, month, day), week, and season, and obtain a total test score ranging from 0 to 10. **Word recall** measured episodic memory. Participants were asked to provide correct immediate and delayed recall of 10 Chinese words, and scores were averaged across the two recall sessions, with test scores ranging from 0 to 10. **Figure Drawing** measured visual-spatial. Participants were asked to draw on paper a picture shown by the investigator of two pentagrams overlapping overlapped, if successful, 1 score was received (otherwise 0). In summary, the overall cognitive score ranged from 0 to 21, the higher the score, the better the cognitive function.

## Covariates

At baseline (2011), the following information was collected face-to-face by systematically trained interviewers using a structured questionnaire: (1) Demographic characteristics (age, sex, living residence, marital status, and education level); (2) Health habits (smoking and drinking status, and social activity); (3) Health status (depressive symptoms, functional disability, and number of chronic diseases); and nighttime sleep duration as confounders in the current study. Age and nighttime sleep duration were treated as continuous variables, while the others (including sex, living residence, marital status, education level, smoking status, drinking status, social activity, depressive symptoms, functional disability, and number of chronic diseases) were categorical variables. Sex was dichotomized as male and female. Living residence was dichotomized as rural and urban. Marital status was dichotomized as married/cohabitated and other (separated, divorced, widowed, and unmarried). Education level was recorded as no formal education, primary school, middle or high school, and college or

above. Smoking status was defined as nonsmoker, light to moderate (<20 cigarettes/day currently or history of smoking), and heavy (≥20 cigarettes/day currently) [15]. Drinking status was defined as nondrinker, light to moderate (<twice a day currently), and heavy (≥twice a day currently) [15]. Social activity was assessed by asking participants whether they had engaged in specific activities in the past month, such as participation in a community-related organization, and was further categorized as: none (<1/month), some (1/month), and active (>1/month). Depressive symptoms were measured using the 10-item Center for Epidemiological Studies Depression Scale (CESD-10), and scores greater than or equal to 12 were considered having depressive symptoms, otherwise not [38]. Additionally, individuals with physician-diagnosed emotional or psychiatric problems were also considered to have depressive symptoms. Functional disability (ranging from 0 to 11) was measured by the number of limitations in activities of daily living (ADLs) and instrumental activities of daily living (IADLs) and further categorized as: none (0 functional disability), mild (1–2 functional disabilities), and severe (>2 functional disabilities). Physician-diagnosed comorbidities were self-reported and included hypertension, dyslipidemia, diabetes, cancer, chronic lung disease, liver disease, heart problems, stroke, kidney disease, digestive disease, memory-related disease, arthritis or rheumatism, and asthma. Number of chronic diseases were further categorized as: 0 (none), 1 (1 chronic disease), and ≥2 (≥2 chronic diseases) [34]. To assess the nighttime sleep duration, participants were asked: During the past month, how many hours of actual sleep did you get at night (average hours for one night)?

## Statistical analyses

Long-term trajectories of cognitive function were determined using group-based trajectory modelling (GBTM), the model that can explore the developmental pattern and heterogeneity of cognitive function over time. The Stata traj plugin was conducted to evaluate the trajectories of cognitive function. The Bayesian information criterion (BIC), an adequate sample size in each group, an average posterior probability (AvePP) value ≥70%, and the interpretability of the model to explain the data were considered to specify the final number of groups and their polynomial functions [26,39]. For the BIC, the lowest absolute values indicate a better model. Therefore, three trajectories model (trajectory shape 2 2 2) was used in subsequent analyses (S2 Table).

Frequency (percentage) was used to describe categorical variables. The normality of continuous variables (age and nighttime sleep duration) was assessed using the Shapiro-Wilk test. The results showed that these variables did not follow a normal distribution ($P<0.05$), therefore, they were presented through median (interquartile range). To compare baseline characteristics of distinct trajectories of cognitive function, $\chi2$ test was used for categorical variables, if continuous variables passed Bartlett's test with ANOVA, otherwise the Kruskal-Wallis $H$ test.  However, as multinomial logistic regression does not require covariates to be normally distributed, this deviation does not affect the validity of the analysis. To examine the association between midday napping and long-term trajectories of cognitive function, the multinomial logistic regression model was used to evaluate risk ratios (RRs) with 95% confidence intervals (CIs). Three models were estimated: model 1, adjusted for age, sex, living residence, marital status, and education level; model 2, adjusted for smoking status, drinking status, social activity, depressive symptoms, functional disability, and number of chronic diseases, plus variables in model 1; model 3, adjusted for nighttime sleep duration, plus variables in model 2. Considering the age and sex differences in the impact of midday napping and cognitive function, we evaluated the association between midday napping and long-term trajectories of cognitive function stratified by age and sex; we also assessed the association whether differed by age (45–59 and ≥60 years) and sex (male and female) by introducing the interaction term into multinomial logistic regression models.

Moreover, participants who all reported midday napping at both baseline (2011) and wave 3 (2015) were used to evaluate the association of longitudinal changes in midday napping with cognitive function. Remained at midday napping 1–30 minutes as reference group. Two sensitivity analyses were performed: (1) after excluding respondents with memory-related diseases at baseline (n=49) to reduce recall bias; and (2) after assuming baseline covariates were missing at random (n=125), and multiple imputation of chained equations was used to create 10 imputed data sets, and the

"mi estimate" command of Stata software was used to pool the results. All analyses were conducted using Stata software (version SE16). $P<0.05$ was set as the threshold for statistical significance (two-tailed).

## Results

Among the 4648 participants, compared with the stable cognitive function trajectory, participants in the slow and rapid cognitive decline trajectories were more likely to be older, male, rural residence, not married/cohabitated, lower education level (no former education and primary school), light to moderate and heavy smoker, heavy drinker, no social activity, had depressive symptoms, mild and severe functional disability, ≥1 chronic diseases, and shorter nighttime sleep duration (Table 1). Moreover, a decreasing cognitive score during follow-up was found (S1 Table).

As shown in Fig 2, three different trajectories of cognitive function reflected patterns of rapid decline, slow decline, and stable over the 4 years. Rapid decline trajectory (n=475, 10.22%) represented participants whose cognitive score decreased rapidly from 2011 to 2013 and slowly from 2013 to 2015. Slow decline trajectory (n=2591, 55.74%) represented participants whose cognitive score decreased slowly from 2011 to 2015. Stable trajectory (n=1582, 34.04%) represented participants whose cognitive score remained at a stable and high level.

The risk of rapid decline was 1.45-fold (95% CI=1.05–2.01) for 0 minutes, 1.49-fold (95% CI=1.05–2.12) for 31–90 minutes, and 2.19-fold (95% CI=1.41–3.42) for >90 minutes, compared with participants reporting midday napping 1–30 minutes. Similar results were found for the risk of slow decline: 1.22-fold (95% CI=1.02–1.47) for 0 minutes, 1.27-fold (95% CI=1.04–1.55) for 31–90 minutes, and 1.80-fold (95% CI=1.38–2.35) for >90 minutes (Table 2). Similar results were found in the sensitivity analyses (S3 Table). The association between midday napping and long-term trajectories of cognitive function did not differ by age (45–59 and ≥60 years) and sex (male and female) (all $P$ value for interaction>0.05) (S4 Table).

We further examined the association between changes (from 2011 to 2015) in midday napping and trajectories of cognitive function. Compared with participants reporting both midday napping 1–30 minutes, the risk of rapid decline was 2.56-fold (95% CI=1.10–5.99) for participants who transferred from >90 minutes to 31–90 minutes, 2.54-fold (95% CI=1.17–5.48) for participants who switched from 31–90 minutes to >90 minutes, and 2.43-fold (95% CI=1.11–5.32) for participants who persisted in >90 minutes. The risk of slow decline was 2.61-fold (95% CI=1.58–4.31) for participants who transferred from >90 minutes to 31–90 minutes, 1.87-fold (95% CI=1.15–3.04) for participants who switched from 31–90 minutes to >90 minutes, and 1.66-fold (95% CI=1.02–2.69) for participants who persisted in >90 minutes (Table 3).

## Discussion

This longitudinal study explored the association between midday napping and the long-term trajectory of cognitive function. Three distinct trajectories of cognitive function were identified and characterized as rapid decline, slow decline, and stable. Individuals with no and long (31–90/>90 minutes) midday napping had an increased likelihood of having trajectories of cognitive decline, especially the rapid decline. Participants who persistently had unhealthy midday napping (transferred from >90–31–90 minutes, switched from 31–90 to >90 minutes, and persisted in >90 minutes) were at increased risk for trajectories of cognitive decline, especially rapid decline.

In line with previous studies, participants with no and long midday napping (>30, >60, or >90 minutes) were at a higher risk of cognitive decline [15,20,21]. A cross-sectional study in older adults found that no and long (>30 minutes) napping was associated with worsing overall cognition [15]. A previous longitudinal study based on CHARLS provided similar evidence and found that no midday napping and extended napping (>90 minutes) were negatively associated with global cognitive function [21]. Prior studies have also shown that napping and brain aging (e.g., Alzheimer's dementia) were linked in a bidirectional relationship, i.e., extended daytime napping increases the risk of dementia in later life, while having Alzheimer's disease leads to an increase in the duration of napping with aging [20]. A strong association between napping for more than 1 hour and the risk of dementia was found during 14 years of follow-up [20]. Similar findings were

**Table 1. Baseline characteristics according to trajectories of cognitive function (n=4648).**

| Characteristics | Total sample (n=4648) | Trajectory Group | | | P [c] |
|---|---|---|---|---|---|
| | | Rapid Decline (n=475) | Slow decline (n=2591) | Stable (n=1582) | |
| Midday napping, minutes | | | | | 0.001 |
| 0 | 1926 (41.44) | 207 (43.58) | 1069 (41.26) | 650 (41.09) | |
| 1-30 | 895 (19.26) | 79 (16.63) | 467 (18.02) | 349 (22.06) | |
| 31-90 | 1321 (28.42) | 134 (28.21) | 739 (28.52) | 448 (28.32) | |
| >90 | 506 (10.89) | 55 (11.58) | 316 (12.20) | 135 (8.53) | |
| Age, median (IQR) [a] | 56.00 (13.00) | 59.00 (13.00) | 57.00 (13.00) | 54.00 (12.00) | <0.001 |
| Male [a] | 2706 (58.24) | 271 (57.17) | 1591 (61.43) | 844 (53.35) | <0.001 |
| Rural residence | 2210 (47.55) | 304 (64.00) | 1322 (51.02) | 584 (36.92) | <0.001 |
| Married/Cohabitated | 4312 (92.77) | 426 (89.68) | 2388 (92.17) | 1498 (94.69) | <0.001 |
| Education level [a] | | | | | <0.001 |
| No formal education | 764 (16.44) | 182 (38.32) | 468 (18.07) | 114 (7.21) | |
| Primary school | 1114 (23.97) | 149 (31.37) | 715 (27.61) | 250 (15.80) | |
| Middle or high school | 2282 (49.11) | 133 (28.00) | 1212 (46.80) | 937 (59.23) | |
| College or above | 487 (10.48) | 11 (2.32) | 195 (7.53) | 281 (17.76) | |
| Smoking status | | | | | <0.001 |
| Nonsmoker | 2574 (55.38) | 246 (51.79) | 1377 (53.15) | 951 (60.11) | |
| Light to moderate | 1002 (21.56) | 131 (27.58) | 570 (22.00) | 301 (19.03) | |
| Heavy | 1072 (23.06) | 98 (20.63) | 644 (24.86) | 330 (20.86) | |
| Drinking status | | | | | <0.001 |
| Nondrinker | 2812 (60.50) | 308 (64.84) | 1533 (59.17) | 971 (61.38) | |
| Light to moderate | 1581 (34.01) | 134 (28.21) | 894 (34.50) | 553 (34.96) | |
| Heavy | 255 (5.49) | 33 (6.95) | 164 (6.33) | 58 (3.67) | |
| Social activity | | | | | <0.001 |
| None | 1817 (39.09) | 230 (48.42) | 1072 (41.37) | 515 (32.55) | |
| Some | 1583 (34.06) | 159 (33.47) | 880 (33.96) | 544 (34.39) | |
| Active | 1248 (26.85) | 86 (18.11) | 639 (24.66) | 523 (33.06) | |
| Depressive symptoms [a,b] | | | | | <0.001 |
| No | 3867 (84.47) | 358 (76.99) | 2108 (82.60) | 1401 (89.75) | |
| Yes | 711 (15.53) | 107 (23.01) | 444 (17.40) | 160 (10.25) | |
| Functional disability [a] | | | | | <0.001 |
| None | 3990 (86.80) | 390 (83.33) | 2187 (85.33) | 1413 (90.23) | |
| Mild | 483 (10.51) | 55 (11.75) | 302 (11.78) | 126 (8.05) | |
| Severe | 124 (2.70) | 23 (4.91) | 74 (2.89) | 27 (1.72) | |
| Number of chronic diseases | | | | | 0.087 |
| 0 | 1691 (36.38) | 165 (34.74) | 908 (35.04) | 618 (39.06) | |
| 1 | 1344 (28.92) | 147 (30.95) | 757 (29.22) | 440 (27.81) | |
| ≥2 | 1613 (34.70) | 163 (34.32) | 926 (35.74) | 524 (33.12) | |
| Nighttime sleep duration, median (IQR), hours [a] | 7.00 (2.00) | 7.00 (3.00) | 7.00 (2.00) | 7.00 (2.00) | 0.034 |

a Missing data: 1 for age, 2 for sex, 1 for education level, 70 for depressive symptoms, 51 for functional disability, and 6 for nighttime sleep duration.

b The 10-item Center for Epidemiological Studies Depression Scale (CESD-10) is greater than or equal to 12.

c Categorical variables were based on $\chi^2$ test and continuous variables were analyzed by ANOVA if they passed Bartlett's test, and otherwise Kruskal-Wallis $H$ test.

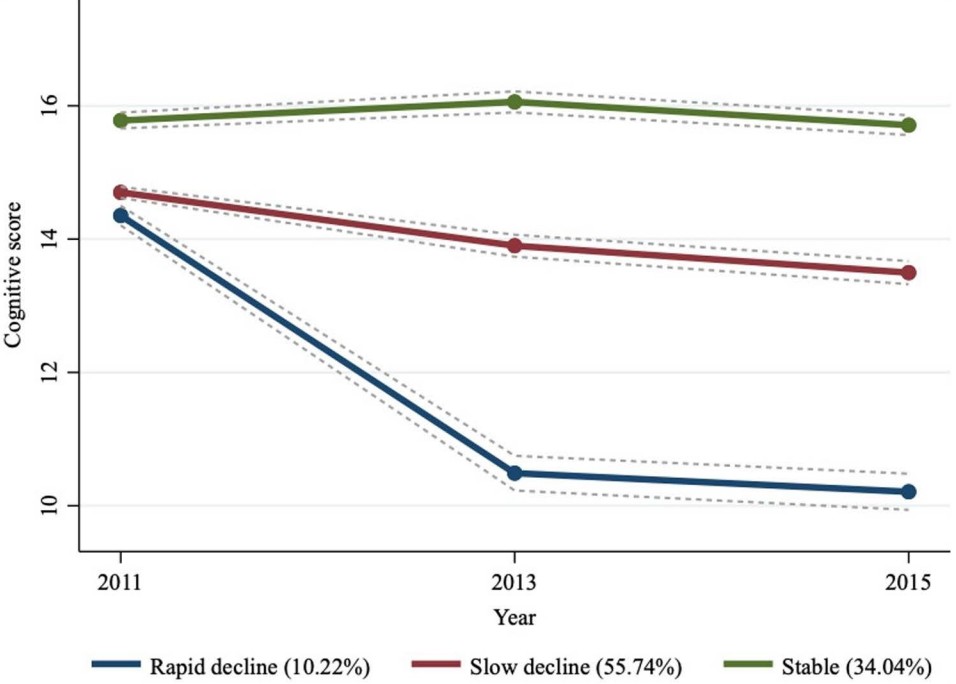

**Fig 2. Trajectory of the cognitive function from 2011 to 2015.** The solid lines display estimated values and the dotted lines represent a 95% confidence interval.

**Table 2. Risk of cognitive decline according to midday napping.**

| Outcome | Case, No. (%) | Risk Ratio (95%CI) | | |
|---|---|---|---|---|
| | | Model 1 | Model 2 | Model 3 |
| **Rapid decline** | | | | |
| Midday napping (minutes) | | | | |
| 0 | 207 (43.58) | 1.36 (1.00-1.86) * | 1.46 (1.05-2.02) * | 1.45 (1.05-2.01) * |
| 1-30 | 79 (16.63) | 1 [Reference] | 1 [Reference] | 1 [Reference] |
| 31-90 | 134 (28.21) | 1.33 (0.96-1.86) | 1.48 (1.04-2.11) * | 1.49 (1.05-2.12) * |
| >90 | 55 (11.58) | 1.85 (1.21-2.82) ** | 2.16 (1.38-3.36) ** | 2.19 (1.41-3.42) ** |
| **Slow decline** | | | | |
| Midday napping (minutes) | | | | |
| 0 | 1069 (41.26) | 1.21 (1.01-1.44) * | 1.22 (1.02-1.47) * | 1.22 (1.02-1.47) * |
| 1-30 | 467 (18.02) | 1 [Reference] | 1 [Reference] | 1 [Reference] |
| 31-90 | 739 (28.52) | 1.21 (1.00-1.46) | 1.27 (1.04-1.55) * | 1.27 (1.04-1.55) * |
| >90 | 316 (12.20) | 1.68 (1.30-2.17) *** | 1.80 (1.38-2.36) *** | 1.80 (1.38-2.35) *** |

Model 1 was adjusted for age, sex, living residence, marital status, and education level.

Model 2 was adjusted for smoking status, drinking status, social activity, depressive symptoms, functional disability, and number of chronic diseases, plus variables in model 1.

Model 3 was adjusted for nighttime sleep duration, plus variables in model 2.

*$P<0.05$, **$P<0.01$, ***$P<0.001$

**Table 3. Association of change in midday napping with trajectories of cognitive function.**

| 2011 | 2015 | Case, No. | Rapid decline | Slow decline |
|------|------|-----------|---------------|--------------|
| | | | RR (95% CI) | RR (95% CI) |
| 0 | 0 | 1060 | 1.68 (0.94-3.00) | 1.23 (0.89-1.72) |
| 1-30 | 0 | 182 | 1.24 (0.57-2.71) | 0.96 (0.61-1.51) |
| 31-90 | 0 | 225 | 1.88 (0.90-3.96) | 1.39 (0.90-2.15) |
| >90 | 0 | 75 | 2.31 (0.85-6.27) | 1.33 (0.71-2.50) |
| 0 | 1-30 | 220 | 0.87 (0.40-1.86) | 0.75 (0.49-1.15) |
| 1-30 | 1-30 | 245 | 1 [Reference] | 1 [Reference] |
| 31-90 | 1-30 | 201 | 1.06 (0.47-2.38) | 1.15 (0.74-1.78) |
| >90 | 1-30 | 55 | 0.83 (0.23-2.97) | 1.09 (0.54-2.18) |
| 0 | 31-90 | 346 | 1.18 (0.59-2.37) | 1.24 (0.84-1.83) |
| 1-30 | 31-90 | 272 | 0.58 (0.27-1.29) | 0.86 (0.58-1.29) |
| 31-90 | 31-90 | 570 | 1.13 (0.60-2.13) | 1.02 (0.71-1.45) |
| >90 | 31-90 | 170 | 2.56 (1.10-5.99) * | 2.61 (1.58-4.31) *** |
| 0 | >90 | 102 | 2.31 (0.96-5.57) | 1.26 (0.72-2.23) |
| 1-30 | >90 | 99 | 1.87 (0.76-4.63) | 1.36 (0.75-2.47) |
| 31-90 | >90 | 185 | 2.54 (1.17-5.48) * | 1.87 (1.15-3.04) * |
| >90 | >90 | 169 | 2.43 (1.11-5.32) * | 1.66 (1.02-2.69) * |

Abbreviations: RR, risk ratio; CI, confidence interval.

Adjusted for age, sex, living residence, marital status, education level, smoking status, drinking status, social activity, depressive symptoms, functional disability, number of chronic diseases, and nighttime sleep duration.

Remained at 1–30 minutes of midday napping in 2011 and 2015 as the reference group.

*P<0.05, ***P<0.001.

found in the present study, i.e., extended napping (>90 minutes) tended to be associated with a higher risk of rapid decline in cognitive function than other napping durations.

Although the relationship between changes in midday napping contributing to cognition has been studied, it remains controversial. Our findings suggest that consistently longer midday napping (persistently >90 minutes, transferred 31–90 to >90 minutes, and switched >90 to31–90 minutes) were detrimental to cognitive function, which was supported by a study based on the Rush Memory and Aging Project [20]. However, this finding was not consistent with another longitudinal study. For example, a previous study found that after controlling for potential confounders, poorer cognitive performance was associated with switching from short midday napping (1–30 minutes) to no and long (>90 minutes) midday napping and from no midday napping to short and long midday napping [40], which was not statistically associated with these findings in the present study. Part of the reason for the difference may be the present study of repeated measures of cognitive function (long-term trajectory of change) and the different study populations (45–59 vs. 60+ years). It also suggests that the risk of switching from short to long periods of cognitive function may be more pronounced in older adults (60+ years) [40].

The underlying mechanism of the association between midday napping and cognition is multifactorial. For example, we considered the relevant biological mechanisms. Research indicates that abnormal accumulation of Tau protein is closely associated with cognitive decline. Tau protein damages brain regions that regulate wakefulness and sleep, potentially leading to circadian rhythm disturbances. These disruptions can increase daytime napping and deteriorate sleep quality, creating a vicious cycle that exacerbates Tau accumulation and further impairs cognitive function [41,42]. It is worthwhile to notice that a normal sleep cycle aids in clearing metabolic byproducts such as β-amyloid and Tau protein from the brain

[43]. Poor sleep quality may contribute to the accumulation of these toxic proteins, thereby affecting cognitive abilities [44]. Chronic inflammation is also a significant factor influencing cognitive function; insufficient sleep can elevate inflammatory markers, which may damage neural cells and their functions [45]. Moreover, the concrete context of education and social development in middle-aged and older Chinese adults is also an important factor. First, they may easily experience sleep deprivation due to work and deterioration of social functions [8,10], which is consistent with a study based on a Chinese population. The study found that the relationship between sleep duration and sleep quality varied by occupation, e.g. blue-collar workers were at a higher risk of experiencing shorter sleep duration and poorer sleep quality, which may be due to differences in socio-economic status [46]. And these sleep disorders may affect sleep recovery beyond two days, further affecting cognitive function [47]. There also seems to be a suggestion that subsequent research could explore the relationship between sleep and cognitive function from the perspective of different occupations. Second, cognitive reserve may be a potential mechanism to prevent sleep-related cognitive impairment, and the degree of cognitive decline correlates with education level [48]. The proportion of middle-aged and older adults with 6 years or above of education was 10.50% in this study, suggesting that they may have a lower level of cognitive reserve and may be better unable to cope with cognitive impairment due to sleep disorders [48]. Third, the risk of cognitive impairment tends to increase with age [9], and China is facing a huge family and social burden of aging [5]. Cognitive function also showed a decline over time in this study. This means that properly dealing with aging and improving their cognitive reserve in terms of sleep disorders and cognitive impairment may be an important component that optimizes cognitive impairment in middle-aged and older adults [6].

It is worthwhile that cognitive function is also influenced by other factors such as physical capability (including handgrip strength and walking speed) [49,50]. Importantly, there may be a bidirectional relationship between them and midday napping, e.g., improved physical capability may enhance the restorative effects of napping, while adequate napping may promote physical recovery by restoring energy and reducing fatigue. This interaction between physical capability and midday napping may have a cumulative effect on cognitive health. Therefore, future research is needed to explore the association between the interaction of napping and physical capability and cognitive function.

This study had several limitations. First, the measure of midday napping was self-reported, which may introduce recall bias in the study. However, the rate of bias was lowest in the Chinese population compared with other races (whites, blacks, and Hispanics) [51], which makes self-reported sleep data more economical and scientifically sound for large-scale population-based studies in China, and increases the utility of our findings for health education to improve sleep habits [52]. Second, the assessment of midday napping in CHARLS is relatively limited, focusing only on basic indicators such as duration. However, previous studies have shown that the timing of napping after lunch and sleep efficiency also influence cognitive function [15,47]. Therefore, we compensated for the lack of such detailed information in CHARLS by including nighttime sleep duration in the statistical analyses. Third, to reduce concerns about reverse causality, i.e., individuals with cognitive impairment may also exhibit sleep disorders. Participants with cognitive score less than or equal to the mean at baseline were excluded. Finally, the sample size was significantly reduced from the initial 17705 participants to the final 4648 (approximately 25% of the original sample) due to the application of exclusion criteria. While these criteria were designed to ensure data quality and homogeneity, this substantial attrition may have introduced selection bias, potentially affecting the generalizability of the findings. Future research should consider the above limitations. Despite these limitations, the strengths of our study are based on a large population-based longitudinal study, and reliable long-term trajectories of cognitive function were found using GBTM. In addition, a wide range of confounders were adjusted, such as depressive symptoms and chronic diseases.

## Conclusions

In summary, this study found three distinct long-term trajectories of cognitive function in Chinese adults (aged 45+ years) and identified no, >30 minutes, transferring from >90 to 31–90 minutes switching from 31–90 to >90 minutes, and

persistent >90 minutes of midday napping as risk factors for cognitive decline, especially rapid decline. In addition, our findings suggested that the trajectory of cognitive function may inform long-term patterns beyond the assessment of cognitive function at a single point in time. To reduce the long-term burden of cognitive impairment and dementia in middle-aged and older Chinese adults, identifying those with the above poor midday napping habits may be an inexpensive and effective approach to screening the population at risk for cognitive decline.

## Supporting information

**S1 Table. Description of cognitive score among the included participants (n=4648) over time from waves 1–3.**
(DOCX)

**S2 Table. Goodness-of-fit statistics of group-based trajectory analysis.**
(DOCX)

**S3 Table. Sensitivity analyses of the risk of cognitive decline according to midday napping.**
(DOCX)

**S4 Table. Risk of cognitive decline according to midday napping stratified by age and sex.**
(DOCX)

## Acknowledgments

We thank the CHARLS research team and participants.

## Author contributions

**Conceptualization:** Jinghong Huang, Xiaohui Wang.

**Formal analysis:** Jinghong Huang, Yutong Zhang, Yanan Zhang.

**Methodology:** Jinghong Huang, Xiaohui Wang.

**Software:** Jinghong Huang.

**Supervision:** Xiaohui Wang.

**Validation:** Yutong Zhang, Yanan Zhang.

**Writing – original draft:** Jinghong Huang.

**Writing – review & editing:** Jinghong Huang, Dongrui Peng, Xiaohui Wang.

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
