## [Decision Letter · Decision Letter 0]

20 May 2024

PONE-D-24-03278Association between midday napping and long-term trajectories of cognitive function among middle-aged and older Chinese adultsPLOS ONE

Dear Dr. Wang,

Thank you for submitting your manuscript to PLOS ONE. After careful consideration, we feel that it has merit but does not fully meet PLOS ONE’s publication criteria as it currently stands. Therefore, we invite you to submit a revised version of the manuscript that addresses the points raised during the review process.

We look forward to receiving your revised manuscript.

Kind regards,

Elvan Wiyarta, M.D.

Academic Editor

PLOS ONE

Journal Requirements:

Reviewers' comments:

Reviewer's Responses to Questions

**Comments to the Author**

1. Is the manuscript technically sound, and do the data support the conclusions?

Reviewer #1: No

Reviewer #2: Partly

2. Has the statistical analysis been performed appropriately and rigorously? 

Reviewer #1: No

Reviewer #2: Yes

3. Have the authors made all data underlying the findings in their manuscript fully available?

Reviewer #1: Yes

Reviewer #2: Yes

4. Is the manuscript presented in an intelligible fashion and written in standard English?

Reviewer #1: Yes

Reviewer #2: Yes

5. Review Comments to the Author

Reviewer #1: the authors need to address on the full time/duration of the night sleep, at least as the covariates in logistical regression. Also, the sleep quality and pattern need to be addressed and discussed. So, it had better re-do the data analysis first.

Reviewer #2: This study was conducted very well with large number of participants and statistic analysis was nicely performed.

However, the introduction was not declaring the title enough about the midday napping itself.

Authors highlighted a rapid and slow cognitive decline, which was very interesting, nevertheless it was not mention at all in the introduction nor methods and how it was calculated statistically. The figure 2 could be more statistically informative. It was also interesting, how the rapid cognitive decline was only found until 2013, was there any explanation?

In the discussion authors mention about unhealthy midday napping, and it only appeared once without prior introduction.

6. PLOS authors have the option to publish the peer review history of their article (what does this mean? ). If published, this will include your full peer review and any attached files.

**Do you want your identity to be public for this peer review?** For information about this choice, including consent withdrawal, please see our Privacy Policy .

Reviewer #1: **Yes: ** wenjie sun

Reviewer #2: **Yes: ** Astri Budikayanti

---

## [Author Response · Author response to Decision Letter 0]

9 Jun 2024

Manuscript title: Association between midday napping and long-term trajectories of cognitive function among middle-aged and older Chinese adults

Submission ID: PONE-D-24-03278

Authors: Jinghong Huang; Dongrui Peng; Xiaohui Wang

Journal: PLOS ONE

Type: Research Paper

Submitted on: Jan 25 2024

Dear Editor,

We gratefully thank you and all reviewers for the constructive remarks and useful suggestions, which have significantly enabled us to improve the manuscript. We are delighted at your decision to invite us to revise and resubmit this manuscript.

The manuscript has been revised according to the comments from the editor and reviewers, and all revisions have been used tracked changes for ready identification. The Point-by-point response to reviewers can be found below. We also prepared the track change and unmarked version for your reference and shared it with editor and reviewers.

We hope this revision is satisfactory and our manuscript will be acceptable for publication in the journal.

Sincerely,

Authors 

Point-by-point response to editor and reviewers

Editor

COMMENTS

Response: Thank you for your comment.

We have carried out a careful check and have met the above requirements.

Response: We gratefully appreciate your careful check.

When we submitted our revisions, we did not see the "Financial Disclosure" sections in the submission system. Therefore, we are stating "Financial Disclosure Statement" here: This study was supported by the National Natural Science Foundation of China (Grant No. 72274088). The funders had no role in study design, data collection and analysis, decision to publish, or preparation of the manuscript.

Response: We gratefully appreciate your careful check.

There are no restrictions.

Data are from the China Health and Retirement Longitudinal Study(CHARLS) and are publicly available from the CHARLS website: https://charls.charlsdata.com/pages/data/111/zh-cn.html. Dataset titles used in this study were 2015 CHARLS Wave 3, 2013 CHARLS Wave2, and 2011 CHARLS Wave1). The present study's authors had no special privileges in accessing these datasets which other interested researchers would not have.

Our URL has no fault, which is consistent with the previous CHARLS study published in PLOS ONE. DOI: 10.1371/journal.pone.0245670; DOI: 10.1371/journal.pone.0226279.

Reviewer 1

COMMENTS

1. the authors need to address on the full time/duration of the night sleep, at least as the covariates in logistical regression. Also, the sleep quality and pattern need to be addressed and discussed. So, it had better re-do the data analysis first.

Response: We gratefully appreciate your constructive suggestion.

Thank you for your constructive comments. Firstly, the nighttime sleep duration was included as a covariate in logistic regression model. However, we missed that variable in the analysis of “Association of change in midday napping over 4-years with cognitive decline risk”. Therefore, for this part of the results, we re-analysed the results after also incorporating nighttime sleep duration as a covariate. See Table 3, which is generally consistent with the previous results.

We similarly considered that midday napping, nighttime sleep duration, and sleep quality are closely related to each other due to circadian rhythms. Therefore, we included nighttime sleep duration and sleep quality-related variables at baseline, in order to examine longitudinal association between midday napping and trajectories of cognitive function. However, the present study had adjusted for depression as a covariate, rather than sleep quality, because:

1. In the CHARLS database, sleep quality is included as part of the assessment of depression (the CESD-10 scale is used to assess depression). 10-items of CESD-10 scale includes: (1) bothered by little things, (2) had trouble concentrating, (3) felt depressed, (4) everything was an effort, (5) felt hopeless, (6) felt fearful, (7) sleep was restless, (8) felt unhappy, (9) felt lonely, and (10) could not get going. Each item was scored as follows: 0 (rarely or none of the time, <1 day), 1 (some or little of the time, 1–2 days), 2 (occasionally or a moderate amount of the time, 3–4 days), and 3 (most or all of the time, 5–7 days). The item “sleep was restless” is always used to assess sleep quality in CHARLS research[1, 2].

2. Depression, as a common psychological factor, may have an impact on the relationship between midday napping and cognitive function. The main points are: (1) Depression may affect cognitive function: patients with depression are often accompanied by impairment of cognitive function, such as diminished attention, memory, and executive function. (2) Depression may affect napping habits: depressed patients may have sleep disorders, such as difficulty in falling asleep and waking up early, and these symptoms may affect patients' midday napping habits. Therefore, adjusting for depression as a covariate can reduce the interference of depression (including the assessment of sleep quality) on the results of the study, thus increasing the sensitivity of the study of the relationship between midday napping and cognitive function, making the results more accurate and reliable.

In summary, we have made improvement in response to your valuable comments:

In the Discussion section, we have added a discussion on sleep quality. The revised text states: For example, a clinical study found that Tau proteins damage brain regions that regulate wakefulness and sleep states, leading to an increase in napping, which affects circadian rhythms and leads to disrupted day and night sleep rhythms and worsen sleep quality, which in turn leads to more Tau proteins and amyloid-like proteins clumping together in the brain, further deteriorating cognitive function[3]. Moreover, combined with the concrete context of education and social development in middle-aged and older Chinese adults. First, they may easily experience sleep deprivation due to work and deterioration of social functions, respectively[4, 5], which is consistent with a study based on a Chinese population. The study found that the relationship between sleep duration and sleep quality varied by occupation, e.g. blue-collar workers were at higher risk of experiencing shorter sleep duration and poorer sleep quality, which may be due to differences in socio-economic status[6]. And these sleep disorder may affect sleep recovery beyond two days, further affecting cognitive function[7]. There also seems to be a suggestion that subsequent research could explore the relationship between sleep and cognitive function from the perspective of different occupations. (Pages 16-17, Lines 203-211)

Reviewer 2

COMMENTS

1. This study was conducted very well with large number of participants and statistic analysis was nicely performed.

However, the introduction was not declaring the title enough about the midday napping itself.

Response: Thanks for your professional review of our study.

We added a statement about midday napping to better respond to the title and bridge the context. The added text state: Midday napping is considered a healthy lifestyle with intervening properties. The prevalence of habitual midday napping ranges from 22% to 69% among older adults worldwide[8-11]. Meanwhile, midday napping is also prevalent in China, as Chinese culture considers it an important part of healthy living[10]. (Page 3, Lines 38-40)

2. Authors highlighted a rapid and slow cognitive decline, which was very interesting, nevertheless it was not mention at all in the introduction nor methods and how it was calculated statistically. The figure 2 could be more statistically informative. It was also interesting, how the rapid cognitive decline was only found until 2013, was there any explanation?

Response: Thank you for your suggestion.

In the introduction section, we add research on trajectories of cognitive function. The added text states: However, these studies assessed cognitive function at a single point in time. This approach does not capture the dynamic process of cognitive function, which may be important because middle-aged and older adults appear to experience different patterns of cognitive function over time. For example, in a study of body mass index and trajectories of cognitive function in 5693 Chinese middle-aged and older adults (aged 45 years+), it was found that individuals tended to show the following three longitudinal patterns of cognitive function, including rapid decline, slow decline, and stable[12]. (Page 4, Lines 50-54)

We had a description of the trajectory methodology and statistical calculations in the Methods section (Page 8, Lines 111-116). In addition, in Supplementary file “S2 Table”, we had statistics related to trajectory determination.

We have added more statistical information to Figure 2, namely the proportion of samples in each trajectory group, which will be consistent with the presentation of previous trajectory-related studies[12]. We believe this will be more helpful to the reader. See Figure 2.

Similar trajectories were also found in a cohort study of body mass index and trajectories of cognitive function[12]. In this study, the rapid decline in cognitive function that occurred in 2011-2013 may have been due to the fact that participants in the rapid decline group were likely to be older, rural residence, not married/cohabitated, lower education level (no former education and primary school), current smoker, no social activity, depressed, severe functional disability, and shorter nighttime sleep duration than slow decline and stable groups. These factors are often considered risk factors for cognitive impairment.

3. In the discussion authors mention about unhealthy midday napping, and it only appeared once without prior introduction.

Response: Thank you for your comment.

Our have added the information about unhealthy midday napping. The revised text states: Participants who persistently had unhealthy midday napping (transferred from >90 to 31-90 minutes, switched from 31-90 to >90 minutes, and persisted in >90 minutes) were at increased risk for trajectories of cognitive decline, especially rapid decline. (Page 15, Lines 179-181)

Reference

1. Li M, Wang N, Dupre ME. Association between the self-reported duration and quality of sleep and cognitive function among middle-aged and older adults in China. Journal of Affective Disorders. 2022;304:20-7. doi: 10.1016/j.jad.2022.02.039. PubMed PMID: WOS:000764263400003.

2. Zhao Y, Hu Y, Smith JP, Strauss J, Yang G. Cohort Profile: The China Health and Retirement Longitudinal Study (CHARLS). International Journal of Epidemiology. 2014;43(1):61-8. doi: 10.1093/ije/dys203. PubMed PMID: WOS:000332341300012.

3. Oh J, Eser RA, Ehrenberg AJ, Morales D, Petersen C, Kudlacek J, et al. Profound degeneration of wake-promoting neurons in Alzheimer's disease. Alzheimers & Dementia. 2019;15(10):1253-63. doi: 10.1016/j.jalz.2019.06.3916. PubMed PMID: WOS:000490148500002.

4. Hendriks S, Peetoom K, Bakker C, van der Flier WM, Papma JM, Koopmans R, et al. Global Prevalence of Young-Onset Dementia A Systematic Review and Meta-analysis. Jama Neurology. 2021;78(9):1080-90. doi: 10.1001/jamaneurol.2021.2161. PubMed PMID: WOS:000676017800004.

5. Lee Y, Chi I, Palinkas LA. Retirement, Leisure Activity Engagement, and Cognition Among Older Adults in the United States. Journal of Aging and Health. 2019;31(7):1212-34. doi: 10.1177/0898264318767030. PubMed PMID: WOS:000474901300009.

6. Sun W, Yu Y, Yuan J, Li C, Liu T, Lin D, et al. Sleep Duration and Quality among Different Occupations-China National Study. Plos One. 2015;10(3). doi: 10.1371/journal.pone.0117700. PubMed PMID: WOS:000351284600015.

7. Laharnar N, Fatek J, Zemann M, Glos M, Lederer K, Suvorov AV, et al. A sleep intervention study comparing effects of sleep restriction and fragmentation on sleep and vigilance and the need for recovery. Physiology & Behavior. 2020;215. doi: 10.1016/j.physbeh.2019.112794. PubMed PMID: WOS:000512221600022.

8. dos Santos AA, Ceolim MF, Iost Pavarini SC, Neri AL, Rampazo MK. Association between sleep disorders and frailty status among elderly. Acta Paulista De Enfermagem. 2014;27(2):120-5. doi: 10.1590/1982-0194201400022. PubMed PMID: WOS:000338330900006.

9. Fang W, Li Z, Wu L, Cao Z, Liang Y, Yang H, et al. Longer habitual afternoon napping is associated with a higher risk for impaired fasting plasma glucose and diabetes mellitus in older adults: results from the Dongfeng-Tongji cohort of retired workers. Sleep Medicine. 2013;14(10):950-4. doi: 10.1016/j.sleep.2013.04.015. PubMed PMID: WOS:000324103600005.

10. Li J, Cacchione PZ, Hodgson N, Riegel B, Keenan BT, Scharf MT, et al. Afternoon Napping and Cognition in Chinese Older Adults: Findings from the China Health and Retirement Longitudinal Study Baseline Assessment. Journal of the American Geriatrics Society. 2017;65(2):373-80. doi: 10.1111/jgs.14368. PubMed PMID: WOS:000397006700024.

11. Pace-Schott EF, Spencer RMC. Age-related changes in the cognitive function of sleep. In: Green AM, Chapman CE, Kalaska JF, Lepore F, editors. Enhancing Performance for Action and Perception: Multisensory Integration, Neuroplasticity and Neuroprosthetics, Pt I. Progress in Brain Research. 1912011. p. 75-89.

12. Zhang W, Chen Y, Chen N. Body mass index and trajectories of the cognition among Chinese middle and old-aged adults. Bmc Geriatrics. 2022;22(1). doi: 10.1186/s12877-022-03301-2. PubMed PMID: WOS:000829153500003.

---

## [Decision Letter · Decision Letter 1]

13 Aug 2024

PONE-D-24-03278R1Association between midday napping and long-term trajectories of cognitive function among middle-aged and older Chinese adultsPLOS ONE

Dear Dr. Wang,

Thank you for submitting your manuscript to PLOS ONE. After careful consideration, we feel that it has merit but does not fully meet PLOS ONE’s publication criteria as it currently stands. Therefore, we invite you to submit a revised version of the manuscript that addresses the points raised during the review process.

We look forward to receiving your revised manuscript.

Kind regards,

Elvan Wiyarta, M.D.

Academic Editor

PLOS ONE

Reviewers' comments:

Reviewer's Responses to Questions

**Comments to the Author**

1. If the authors have adequately addressed your comments raised in a previous round of review and you feel that this manuscript is now acceptable for publication, you may indicate that here to bypass the “Comments to the Author” section, enter your conflict of interest statement in the “Confidential to Editor” section, and submit your "Accept" recommendation.

Reviewer #1: (No Response)

2. Is the manuscript technically sound, and do the data support the conclusions?

Reviewer #1: (No Response)

3. Has the statistical analysis been performed appropriately and rigorously? 

Reviewer #1: (No Response)

4. Have the authors made all data underlying the findings in their manuscript fully available?

Reviewer #1: (No Response)

5. Is the manuscript presented in an intelligible fashion and written in standard English?

Reviewer #1: (No Response)

6. Review Comments to the Author

**Reviewer #1:**  The authors partial answered some questions. However some essential questions not answer yet. It is not very difficult to find some significant results in this big dataset. The key is providing the convinced evidences not only from your results but also from biological mechanism. otherwise, it is more like "fishing" game. Logistic regression may not be applied here, because the distribution of the key factors. Also, mental health factors may not works as confounder but as the mediator or moderator et al. Authors need further discuss it, if they refuse re-do the data analysis.

although similar studies have been conduct within this data set, the manuscript did not cited any of them.

7. PLOS authors have the option to publish the peer review history of their article (what does this mean? ). If published, this will include your full peer review and any attached files.

**Do you want your identity to be public for this peer review?** For information about this choice, including consent withdrawal, please see our Privacy Policy .

Reviewer #1: No

---

## [Author Response · Author response to Decision Letter 1]

16 Aug 2024

Point-by-point response to editor and reviewers

Reviewer 1

COMMENTS

1. The authors partial answered some questions. However some essential questions not answer yet. It is not very difficult to find some significant results in this big dataset. The key is providing the convinced evidences not only from your results but also from biological mechanism. otherwise, it is more like "fishing" game. Logistic regression may not be applied here, because the distribution of the key factors. Also, mental health factors may not works as confounder but as the mediator or moderator et al. Authors need further discuss it, if they refuse re-do the data analysis.

Response: We gratefully appreciate your constructive suggestion.

We agreed to re-analyze the data. In the reanalysis, the independent and dependent variables remained unchanged, but among the covariates, the following variables were considered: age, sex, living residence, marital status, education level, smoking status, drinking status, social activity, functional disability, number of chronic diseases, sleep quality, and nighttime sleep duration. Of these, age and nighttime sleep duration were continuous variables and the rest were categorical. We have updated and checked all the information in detail and errata.

We also thank you for your suggestion that ‘mental health factors may not act as confounders, but rather as moderators or mediators’. While we agree that the possible moderating role of mental health factors in the relationship between sleep and cognitive functioning is interesting and worthy of further exploration, the primary goal of this study was to directly examine the relationship between nap duration and long-term trajectories of cognitive functioning.

Therefore, in the new analyses, we included sleep quality as an independent variable in the model, rather than indirectly considering it through the total score of the depression scale CESD-10. This was done to ensure the independence and accuracy of the sleep quality variable in the analyses. We extracted information about sleep quality independently from the questionnaire (DC015. My sleep was restless), which allowed us to examine more explicitly the relationship between nap time and trajectories of cognitive functioning.

This adjustment will help us to understand the relationship between napping and cognitive functioning trajectories more clearly and ensure the scientific validity and reliability of the results. 

Response to the Comment Regarding logistic regression may not be applied here, because the distribution of the key factors:

(1) Suitability of the Dependent Variable: In this study, a Multinomial Logistic Regression Model was employed, which necessitates that the dependent variable is a discrete categorical variable. Our study meets this requirement, with the dependent variable classified into three categories: Rapid Decline Trajectory, Slow Decline Trajectory, and Stable Trajectory. The Stable Trajectory serves as the reference group. The categorization of these groups aligns with the requirements for applying multinomial logistic regression.

(2) Category Balance and Sample Size: For each category of the dependent variable, we ensured an adequate sample size to mitigate potential model instability. Specifically:

Rapid decline trajectory (n=475, 10.22%)

Slow decline trajectory (n=2591, 55.74%)

Stable trajectory (n=1582, 34.04%)

The above sample size ensures that each category is represented in the model and reduces model instability. In addition, the categorical variables in the independent variables were also checked to ensure that each category had sufficient sample size to support the stability and accuracy of the model. See the updated Table 1 for details. 

Thank you for your insightful comment regarding the need for convincing biological evidence in addition to our statistical findings. In response, we have expanded our discussion of the underlying biological mechanisms to elucidate the relationship between midday napping and cognitive function. Now it reads:

“For example, we considered the relevant biological mechanisms. Research indicates that abnormal accumulation of Tau protein is closely associated with cognitive decline. Tau protein damages brain regions that regulate wakefulness and sleep, potentially leading to circadian rhythm disturbances. These disruptions can increase daytime napping and deteriorate sleep quality, creating a vicious cycle that exacerbates Tau accumulation and further impairs cognitive function[1, 2]. It is worthwhile to notice that a normal sleep cycle aids in clearing metabolic byproducts such as β-amyloid and Tau protein from the brain[3]. Poor sleep quality may contribute to the accumulation of these toxic proteins, thereby affecting cognitive abilities[4]. Chronic inflammation is also a significant factor influencing cognitive function; insufficient sleep can elevate inflammatory markers, which may damage neural cells and their functions[5].” (Page 18, Lines 500-507)

By integrating these biological insights, we aim to provide a more comprehensive understanding of how midday napping might influence cognitive function through various physiological pathways. this expanded discussion enhances the credibility of our findings by linking them to well-established biological processes. 

In response to the reviewer’s comment, "although similar studies have been conducted within this dataset, the manuscript did not cite any of them," we would like to provide the following clarification:

We appreciate the reviewer’s observation regarding the limited citation of previous studies utilizing the CHARLS database to explore the relationship between napping and cognitive function. In our original manuscript, we referenced two relevant studies:

(1) Afternoon Napping and Cognition in Chinese Older Adults: Findings from the China Health and Retirement Longitudinal Study Baseline Assessment. Journal of the American Geriatrics Society. 2017; 65(2): 373-80. doi: 10.1111/jgs.14368. PubMed PMID: WOS:000397006700024.

(2) Longitudinal associations between daytime napping and cognitive function in Chinese older adults. Archives of Gerontology and Geriatrics. 2023;107. doi: 10.1016/j.archger.2022.104909. PubMed PMID: WOS:000914099300001.

However, we recognize that the references provided may not fully encompass the scope of existing research in this area. In response to your feedback, we have revised the introduction to more comprehensively summarize and cite previous CHARLS-based studies on this topic. The revised content now includes a broader discussion of the methodological limitations identified in prior research and explains the rationale behind our chosen approach:

“Importantly, while previous CHARLS-based studies have explored the relationship between napping and cognitive functioning both cross-sectionally[6-9] and longitudinally[10], these studies exhibit some methodological limitations. The cross-sectional evidence is limited by the design, and long-term effects are needed to be identified. Limited longitudinal research predominantly relied on the linear mixed-effect model. Although the model accounts for variability in baseline cognitive function and time trends across individuals, they may not capture the full range of cognitive development trajectories within different subgroups. In contrast, the group-based trajectory model provides a method to estimate probabilities for multiple developmental trajectories simultaneously, rather than merely modeling the average effect for the study population[11]. This approach is particularly useful for identifying distinct long-term patterns of cognitive function, which is crucial given that cognitive trajectories in middle-aged and older adults can vary significantly over time.” (Page 4, Lines 85-92)

Additionally, we have streamlined the discussion section by removing a paragraph that was redundant with the revised introduction. To further strengthen our discussion, we have also incorporated additional studies based on the CHARLS database to contextualize our findings within the broader literature. See Page 17, Lines 467-468. 

Reference

1. Lebouvier T, Pasquier F, Buee L. Update on tauopathies. Current Opinion in Neurology. 2017;30(6):589-98. doi: 10.1097/wco.0000000000000502. PubMed PMID: WOS:000415103100006.

2. Wang C, Holtzman DM. Bidirectional relationship between sleep and Alzheimer's disease: role of amyloid, tau, and other factors. Neuropsychopharmacology. 2020;45(1):104-20. doi: 10.1038/s41386-019-0478-5. PubMed PMID: WOS:000499141800010.

3. Xie L, Kang H, Xu Q, Chen MJ, Liao Y, Thiyagarajan M, et al. Sleep Drives Metabolite Clearance from the Adult Brain. Science. 2013;342(6156):373-7. doi: 10.1126/science.1241224. PubMed PMID: WOS:000325755100049.

4. Bojarskaite L, Vallet A, Bjornstad DM, Binder KMG, Cunen C, Heuser K, et al. Sleep cycle-dependent vascular dynamics in male mice and the predicted effects on perivascular cerebrospinal fluid flow and solute transport. Nature Communications. 2023;14(1). doi: 10.1038/s41467-023-36643-5. PubMed PMID: WOS:001164830900022.

5. Everard A, Cani PD. Diabetes, obesity and gut microbiota. Best Practice & Research Clinical Gastroenterology. 2013;27(1):73-83. doi: 10.1016/j.bpg.2013.03.007. PubMed PMID: WOS:000322352400008.

6. Zhang H, Zhang L, Chen C, Zhong X. Association between daytime napping and cognitive impairment among Chinese older population: a cross-sectional study. Environmental Health and Preventive Medicine. 2023;28. doi: 10.1265/ehpm.23-00031. PubMed PMID: WOS:001110201500004.

7. Hu M, Shu X, Feng H, Xiao LD. Sleep, inflammation and cognitive function in middle-aged and older adults: A population-based study. Journal of Affective Disorders. 2021;284:120-5. doi: 10.1016/j.jad.2021.02.013. PubMed PMID: WOS:000623621400015.

8. Zhou L, Zhang Y, Ge M, Zhang G, Cheng R, Liu Y, et al. The associations of daytime napping and motoric cognitive risk syndrome: Findings from the China Health and Retirement Longitudinal Study. Experimental Gerontology. 2024;191. doi: 10.1016/j.exger.2024.112426. PubMed PMID: WOS:001232161800001.

9. Li J, Cacchione PZ, Hodgson N, Riegel B, Keenan BT, Scharf MT, et al. Afternoon Napping and Cognition in Chinese Older Adults: Findings from the China Health and Retirement Longitudinal Study Baseline Assessment. Journal of the American Geriatrics Society. 2017;65(2):373-80. doi: 10.1111/jgs.14368. PubMed PMID: WOS:000397006700024.

10. Zhang L, Chen C, Zhang H, Peng B. Longitudinal associations between daytime napping and cognitive function in Chinese older adults. Archives of Gerontology and Geriatrics. 2023;107. doi: 10.1016/j.archger.2022.104909. PubMed PMID: WOS:000914099300001.

11. Nagin DS, Odgers CL. Group-Based Trajectory Modeling in Clinical Research. In: NolenHoeksema S, Cannon TD, Widiger T, editors. Annual Review of Clinical Psychology, Vol 6. Annual Review of Clinical Psychology. 62010. p. 109-38.

---

## [Decision Letter · Decision Letter 2]

28 Nov 2024

PONE-D-24-03278R2Association between midday napping and long-term trajectories of cognitive function among middle-aged and older Chinese adultsPLOS ONE

Dear Dr. Wang,

Thank you for submitting your manuscript to PLOS ONE. After careful consideration, we feel that it has merit but does not fully meet PLOS ONE’s publication criteria as it currently stands. Therefore, we invite you to submit a revised version of the manuscript that addresses the points raised during the review process.

We look forward to receiving your revised manuscript.

Kind regards,

Elvan Wiyarta, M.D.

Academic Editor

PLOS ONE

Journal Requirements:

Reviewers' comments:

Reviewer's Responses to Questions

**Comments to the Author**

1. If the authors have adequately addressed your comments raised in a previous round of review and you feel that this manuscript is now acceptable for publication, you may indicate that here to bypass the “Comments to the Author” section, enter your conflict of interest statement in the “Confidential to Editor” section, and submit your "Accept" recommendation.

Reviewer #3: All comments have been addressed

Reviewer #4: (No Response)

Reviewer #5: (No Response)

2. Is the manuscript technically sound, and do the data support the conclusions?

Reviewer #3: Yes

Reviewer #4: (No Response)

Reviewer #5: Yes

3. Has the statistical analysis been performed appropriately and rigorously? 

Reviewer #3: Yes

Reviewer #4: (No Response)

Reviewer #5: Yes

4. Have the authors made all data underlying the findings in their manuscript fully available?

Reviewer #3: Yes

Reviewer #4: (No Response)

Reviewer #5: Yes

5. Is the manuscript presented in an intelligible fashion and written in standard English?

Reviewer #3: Yes

Reviewer #4: (No Response)

Reviewer #5: Yes

6. Review Comments to the Author

Reviewer #3: I commend the authors for their thorough revisions and thoughtful responses to the reviewers' comments. The manuscript is significantly improved, and the revisions strengthen its scientific rigor.

Data Reanalysis and Statistical Rigor: The reanalysis of the data, with additional covariates included, is well-executed. The use of multinomial logistic regression is well-justified, and the sample size balance across trajectory groups enhances the robustness of the findings.

Biological Mechanism Discussion: The expanded discussion of biological mechanisms, including Tau protein accumulation and inflammation, strengthens the manuscript and provides a solid biological basis for the observed relationships.

Literature Review: The authors have improved the introduction by including relevant studies from the CHARLS database, providing important context for their findings and justifying their methodology.

Mental Health Factors: The inclusion of sleep quality as an independent variable adds clarity to the analysis and strengthens the validity of the conclusions.

Reviewer #4: This study explores the association between midday napping and long-term trajectories of cognitive function in middle-aged and older Chinese adults. This study may be helpful in screening the population at risk for cognitive decline. In my opinion, the research paper provides an interesting topic, but there are several issues to be addressed as follows:

1. The study initially targeted a group of over 17705 people, which at first glance appears to be a large sample. However, the actual number of participants that remained in the final analysis was only around 4648, which is merely about 25% of the original group. The almost majority of the original participants cannot be used for this study, indicating the potential for significant bias. Despite this, there is no mention of any discussion or acknowledgment of this limitation in the text.

2. MMSE and TICS-10 are two tools to assess cognitive function. However in this work, the authors stated this study used MMSE, and then used TICS-10. Readers will be confused by these.

3. Despite the large sample size, it is recommended to perform and report in the text a normality test of the distribution (example: Shapiro-Wilk).

4. In the introduction, the authors provide information about the association between sleep and cognitive function. In my opinion, one aspect you could introduce into the introduction of you paper is the relationship between sleep, physical capability, and cognitive function. The article (PMID: 37084959; 38835194) may be helpful in this section. In this way, the introduction will be more comprehensive.

5. It is important to determine the relationship according to sex and age group, since sex and age difference (e.g.: middle-aged, young-old, and old-old) may be existed in the relationship.

Reviewer #5: I’m not sure why the authors are masking the impact of emotional or psychiatric problems by lumping them in with chronic physical diseases. This should be a separate variable, particularly as it can affect falling to sleep and sleep quality, and cognitive function, regardless of a person’s physical health problems. (Brown & Mutambudzi, 2023, Aging & Mental Health, 27(6), 1095–1102). Furthermore, mental health factors – emotional and psychiatric problems and CES-D depressive symptoms – have a separate relationship to cognitive function, and should therefore be considered as separate measures in the models. However, the authors are not including CES-D scores in their models, even though their response to the previous review implies that they had initially included this measure. I think excluding depression is a mistake, because it definitely impacts sleep.

The authors do not clearly state they type of variable (dichotomous/nominal, ordinal, continuous/interval) for each covariate they are defining on pages 7-8. For example, they never defined their age variable – is it continuous or are they looking at age groups?

How are they dealing with missingness in these variables?

Drinking status – what are they doing with people who never drink? Aren’t they substantively different from those who drink occasionally? And on what basis have they chosen to define “not occasional drinking” as more than or equal to 4-6 times a week? Is that just the way the categories are defined in the survey?

Please state clearly which variable is your dependent variable – is it midday napping, or is it cognitive function? When you introduce your measures (line 83), you are not labeling these 2 measures as either dependent or independent variables in your models. Don’t make the reader wait until your statistical analyses.

7. PLOS authors have the option to publish the peer review history of their article (what does this mean? ). If published, this will include your full peer review and any attached files.

**Do you want your identity to be public for this peer review?** For information about this choice, including consent withdrawal, please see our Privacy Policy .

Reviewer #3: No

Reviewer #4: No

Reviewer #5: No

---

## [Author Response · Author response to Decision Letter 2]

9 Dec 2024

Point-by-point response to reviewers

Reviewer 3

COMMENTS

1. I commend the authors for their thorough revisions and thoughtful responses to the reviewers' comments. The manuscript is significantly improved, and the revisions strengthen its scientific rigor.

Data Reanalysis and Statistical Rigor: The reanalysis of the data, with additional covariates included, is well-executed. The use of multinomial logistic regression is well-justified, and the sample size balance across trajectory groups enhances the robustness of the findings.

Biological Mechanism Discussion: The expanded discussion of biological mechanisms, including Tau protein accumulation and inflammation, strengthens the manuscript and provides a solid biological basis for the observed relationships.

Literature Review: The authors have improved the introduction by including relevant studies from the CHARLS database, providing important context for their findings and justifying their methodology.

Mental Health Factors: The inclusion of sleep quality as an independent variable adds clarity to the analysis and strengthens the validity of the conclusions.

Response: We are pleased to know that the revised manuscript has improved in scientific rigor and overall quality. Thank you very much for your positive feedback. 

Reviewer 4

COMMENTS

1. The study initially targeted a group of over 17705 people, which at first glance appears to be a large sample. However, the actual number of participants that remained in the final analysis was only around 4648, which is merely about 25% of the original group. The almost majority of the original participants cannot be used for this study, indicating the potential for significant bias. Despite this, there is no mention of any discussion or acknowledgment of this limitation in the text.

Response: Thank you for your insightful comment.

We acknowledge that this substantial attrition (approximately 25% of the original sample) represents a limitation of the study and may have introduced selection bias, potentially affecting the generalizability of the findings. To address this, we have revised the "Limitations" section of the manuscript to explicitly acknowledge this issue and its potential implications. Specifically, we have added the following text: Finally, the sample size was significantly reduced from the initial 17705 participants to the final 4648 (approximately 25% of the original sample) due to the application of exclusion criteria. While these criteria were designed to ensure data quality and homogeneity, this substantial attrition may have introduced selection bias, potentially affecting the generalizability of the findings. Future research should consider the above limitations. (Page 20, Lines 704-707)

2. MMSE and TICS-10 are two tools to assess cognitive function. However in this work, the authors stated this study used MMSE, and then used TICS-10. Readers will be confused by these.

Response: We gratefully appreciate your constructive suggestion. To ensure clarity and avoid confusion, we have revised the manuscript accordingly. Specifically, the text now reads: The dependent variable in this study was long-term trajectories of cognitive function. Cognitive function represented the three ability assessments: orientation, attention, episodic memory, and visual-spatial, measured at baseline and at 2 subsequent follow-up visits (2013 and 2015). The cognitive function measures in CHARLS were derived from the components of the Telephone Interview for Cognitive Status battery (TICS-10), which specifically assessed orientation and attention[1]. (Pages 6-7, Lines 90-101)

We believe this revised description accurately reflects the methodology and eliminates any ambiguity for the readers. Thank you for pointing out this important detail.

3. Despite the large sample size, it is recommended to perform and report in the text a normality test of the distribution (example: Shapiro-Wilk).

Response: We appreciate your attention to the robustness of the statistical analysis.

Among the covariates included in this study, only age and nighttime sleep duration are continuous variables. In response to your suggestion, we conducted Shapiro-Wilk tests for these two variables. The results indicated that both age and nighttime sleep duration did not strictly follow a normal distribution (P<0.05). However, the primary statistical method employed in this study—multinomial logistic regression—does not assume normality of the covariates. Therefore, this deviation from normality does not affect the validity of our findings.

To clarify this, we have added the following statement to the Methods section of the revised manuscript: The normality of continuous variables (age and nighttime sleep duration) was assessed using the Shapiro-Wilk test. The results showed that these variables did not follow a normal distribution (P<0.05). However, as multinomial logistic regression does not require covariates to be normally distributed, this deviation does not affect the validity of the analysis. (Page 9, Lines 156-158)

We hope this revision adequately addresses your concern and enhances the methodological clarity of our manuscript. Thank you again for your valuable feedback.

4. In the introduction, the authors provide information about the association between sleep and cognitive function. In my opinion, one aspect you could introduce into the introduction of you paper is the relationship between sleep, physical capability, and cognitive function. The article (PMID: 37084959; 38835194) may be helpful in this section. In this way, the introduction will be more comprehensive.

Response: We thank your insightful suggestion.

While the primary focus of our study is the association between midday napping and long-term cognitive function trajectories, we agree that considering the role of physical capability could provide a broader perspective on the factors influencing cognitive health. To address this, we have incorporated a forward-looking discussion in the manuscript to highlight this point: It is worthwhile that cognitive function is also influenced by other factors such as physical capability (including handgrip strength and walking speed)[2, 3]. Importantly, there may be a bidirectional relationship between them and midday napping, e.g., improved physical capability may enhance the restorative effects of napping, while adequate napping may promote physical recovery by restoring energy and reducing fatigue. This interaction between physical ability and midday napping may have a cumulative effect on cognitive health. Therefore, future research is needed to explore the association between the interaction of napping and physical ability and cognitive function. (Pages 19, Lines 692-696)

This addition has been placed in the discussion section to ensure the flow of the introduction remains focused on the study objectives. By incorporating this perspective, we aim to underscore the complexity of factors influencing cognitive function and provide directions for future research. We sincerely thank the reviewer for this valuable suggestion, which has enriched the scope of our discussion.

5. It is important to determine the relationship according to sex and age group, since sex and age difference (e.g.: middle-aged, young-old, and old-old) may be existed in the relationship.

Response: Thank you for highlighting the importance of examining potential differences in the relationship between midday napping and cognitive function trajectories across age and sex groups.

In response to your suggestion, we conducted stratified analyses and interaction tests by both age (45-59 and ≥60 years) and sex (male and female). As shown in S4 Table, the results indicate that the differences in the relationship between midday napping and long-term trajectories of cognitive function across these subgroups are not statistically significant (all P value for interaction>0.05).

To ensure clarity and rigor, we have described the stratified analyses and interaction testing in detail in the Methods section (Page 9, Lines 162-164), and the findings have been incorporated into the Supporting Information. These results suggest that while there may be variability in the observed associations, the differences by age and sex are not statistically significant.

Thanks again for your insightful suggestion, which has allowed us to more comprehensively address potential modifiers of the relationship in our study. 

Reviewer 5

COMMENTS

1. I’m not sure why the authors are masking the impact of emotional or psychiatric problems by lumping them in with chronic physical diseases. This should be a separate variable, particularly as it can affect falling to sleep and sleep quality, and cognitive function, regardless of a person’s physical health problems. (Brown & Mutambudzi, 2023, Aging & Mental Health, 27(6), 1095–1102). Furthermore, mental health factors – emotional and psychiatric problems and CES-D depressive symptoms – have a separate relationship to cognitive function, and should therefore be considered as separate measures in the models. However, the authors are not including CES-D scores in their models, even though their response to the previous review implies that they had initially included this measure. I think excluding depression is a mistake, because it definitely impacts sleep.

Response: Thank you for your valuable comments.

We have made some clarifications and adjustments regarding the issue of grouping emotional or psychiatric problems with chronic physical diseases.

In the CHARLS database, emotional or psychiatric problems are included alongside 13 other chronic diseases (such as hypertension, diabetes, etc.) as self-reported conditions. To maintain consistency with the database, we initially listed them together as covariates in our model. However, we understand your concern that emotional or psychiatric problems may independently impact outcomes such as sleep quality and cognitive function, and thus combining them with other chronic diseases may mask their unique effects.

In response, we attempt to treat emotional or psychiatric problems as a separate variable. Specifically, individuals with emotional or psychiatric problems are classified as having depressive symptoms. This approach allows us to account for the independent effects of emotional disorders on the outcomes and ensures their analysis. Consequently, we have removed emotional or psychiatric problems from the list of number of chronic diseases in the covariates to avoid duplication.

We will update the methods section as follows: Physician-diagnosed comorbidities were self-reported and included hypertension, dyslipidemia, diabetes, cancer, chronic lung disease, liver disease, heart problems, stroke, kidney disease, digestive disease, memory-related disease, arthritis or rheumatism, and asthma. (Page 8, Lines128-129)

Summarily, in the updated analysis, emotional or psychiatric problems, together with depressive symptoms measured using the CESD-10 scale were treated as a separate variable and were not included in the number of chronic diseases to avoid duplication. Specifically, the text now reads: Depressive symptoms were measured using the 10-item Center for Epidemiological Studies Depression Scale (CESD-10), and scores greater than or equal to 12 were considered having depressive symptoms, otherwise not[4]. Additionally, individuals with physician-diagnosed emotional or psychiatric problems were also considered to have depressive symptoms. (Page 8, Lines 123-125)

Moreover, we would like to provide clarification regarding our approach to mental health factors and sleep quality in the models.

We acknowledge your point that mental health factors, such as depressive symptoms (CESD-10), should be considered as separate measures in the models. In response, we have carefully reconsidered the role of psychological factors in our analysis and have decided to revert and update to our original approach, where depressive symptoms will be included as a covariate in our model. This decision was driven by several key considerations:

1. Consistent and Updated with Previous Analysis: In our initial analysis, we considered depressive symptoms measured using the CESD-10 scale as a separate covariate because the CESD-10 scale includes a sleep quality-related item (DC015: "My sleep was restless"). Including depressive symptoms as a covariate allows us to better capture the broader impact of psychological well-being on cognitive functioning while avoiding redundancy between the sleep quality item and depressive symptoms. Meanwhile, to comprehensively measure mental health factors, emotional or psychiatric problems, together with depressive symptoms measured using the CESD-10 scale were treated as a separate variable.

2. Avoiding Redundancy and Overlap: The sleep quality question (DC015) is part of the CESD-10 scale, which makes it inherently linked to depressive symptoms. Given this overlap, we felt that keeping the sleep quality item (DC015) in the model alongside depressive symptoms would risk introducing multicollinearity. Hence, we decided to use depressive symptoms as a comprehensive measure to reflect both sleep quality and depressive symptoms in a unified way.

3. Addressing Pre-Reviewer Feedback: While the suggestion to consider mental health factors as confounders rather than mediators is valuable, we consider that depressive symptoms serve as a well-established and comprehensive measure of psychological distress, which justifies its inclusion in the model. This inclusion is consistent with prior research that has linked depressive symptoms to cognitive functioning and underscores the importance of mental health in our analysis.

We are aware that this adjustment may appear as a shift in our analysis strategy. However, we would like to emphasize that our approach is rooted in the goal of providing the most scientifically rigorous and accurate model. After thoroughly considering your comment on mental health as a covariate, we believe that reverting to our original analysis is the most methodologically sound decision. We hope that this rationale clarifies our decision and we welcome any further suggestions from the reviewer.

In our updated analysis, therefore, the independent variable (X=midday napping) remains unchanged, while the dependent variable continues to be trajectories of cognitive function. The following covariates were included in the model: age, sex, living residence, marital status, education level, smoking status, drinking status, social activity, depressive symptoms, functional disability, number of chronic diseases, and nighttime sleep duration. Age and nighttime sleep duration were treated as continuous variables, while the others were categorical.

Thanks again for your suggestion, which has helped us further refine our analysis.

2. The authors do not clearly state they type of variable (dichotomous/nominal, ordinal, continuous/interval) for each covariate they are defining on pages 7-8. For example, they never defined their age variable – is it continuous or are they looking at age groups?

Response: We sincerely appreciate your valuable comment.

We apologize for any confusion regarding the definition of the variables. To clarify, the types of variables included in our models are as follows:

Age and nighttime sleep duration were treated as continuous variables.

All other covariates (including sex, living residence, marital status, education level, smoking status, drinking status, social activity, depressive symptoms, functional disability, and number of chronic diseases) were treated as categorical variables.

Specifically, the text now reads: Age and nighttime sleep duration were treated as continuous variables, while the others (including sex, living residence, marital status, education level, smoking status, drinking status, social activity, depressive symptoms, functional disability, and number of chronic diseases) were categorical. (Page 7, Lines 111-113)

3. How are they dealing with missingness in these variables?

Response: Thanks for your constructive suggestion.

In our main analysis, we did not exclude cases with missing covariates. However, we utilized multiple imputation to address missingness in the covaria

---

## [Decision Letter · Decision Letter 3]

12 Jan 2025

Association between midday napping and long-term trajectories of cognitive function among middle-aged and older Chinese adults

PONE-D-24-03278R3

Dear Dr. Wang,

We’re pleased to inform you that your manuscript has been judged scientifically suitable for publication and will be formally accepted for publication once it meets all outstanding technical requirements.

Kind regards,

Elvan Wiyarta, M.D.

Academic Editor

PLOS ONE

Additional Editor Comments (optional):

Reviewers' comments:

Reviewer's Responses to Questions

**Comments to the Author**

1. If the authors have adequately addressed your comments raised in a previous round of review and you feel that this manuscript is now acceptable for publication, you may indicate that here to bypass the “Comments to the Author” section, enter your conflict of interest statement in the “Confidential to Editor” section, and submit your "Accept" recommendation.

Reviewer #4: All comments have been addressed

2. Is the manuscript technically sound, and do the data support the conclusions?

Reviewer #4: Yes

3. Has the statistical analysis been performed appropriately and rigorously? 

Reviewer #4: Yes

4. Have the authors made all data underlying the findings in their manuscript fully available?

Reviewer #4: Yes

5. Is the manuscript presented in an intelligible fashion and written in standard English?

Reviewer #4: Yes

6. Review Comments to the Author

Reviewer #4: Thank you for revising the manuscript. All my concerns have been adequately addressed. I have no further comments.

7. PLOS authors have the option to publish the peer review history of their article (what does this mean? ). If published, this will include your full peer review and any attached files.

**Do you want your identity to be public for this peer review?** For information about this choice, including consent withdrawal, please see our Privacy Policy .

Reviewer #4: No

---

## [Editor Report · Acceptance letter]

PONE-D-24-03278R3

PLOS ONE

Dear Dr. Wang,

I'm pleased to inform you that your manuscript has been deemed suitable for publication in PLOS ONE. Congratulations! Your manuscript is now being handed over to our production team.

Kind regards,

on behalf of

Mr. Elvan Wiyarta

Academic Editor

PLOS ONE